# Sparstolonin B Suppresses Proliferation and Modulates Toll-like Receptor Signaling and Inflammatory Pathways in Human Colorectal Cancer Cells

**DOI:** 10.3390/ph18030300

**Published:** 2025-02-21

**Authors:** Bürke Çırçırlı, Çağatay Yılmaz, Tuğçe Çeker, Zerrin Barut, Esma Kırımlıoğlu, Mutay Aslan

**Affiliations:** 1Department of Medical Biotechnology, Institute of Health Sciences, Akdeniz University, Antalya 07070, Turkey; burkecircirli@outlook.com; 2Department of Medical Biochemistry, Faculty of Medicine, Akdeniz University, Antalya 07070, Turkey; ccagatayyilmaz@gmail.com (Ç.Y.); tugceker159@gmail.com (T.Ç.); 3Faculty of Dentistry, Antalya Bilim University, Antalya 07190, Turkey; zerrin.barut@antalya.edu.tr; 4Department of Histology and Embryology, Faculty of Medicine, Akdeniz University, Antalya 07070, Turkey; esmakirimlioglu@gmail.com

**Keywords:** Sparstolonin B, colorectal cancer, apoptosis, toll-like receptor, inflammation

## Abstract

**Background:** Sparstolonin B (SsnB), a natural compound with anti-inflammatory and anti-proliferative properties, was investigated for its effects on cell viability, apoptosis, and inflammatory pathways in human colorectal cancer cells (HCT-116) and healthy human fibroblasts (BJ). Phorbol 12-myristate 13-acetate (PMA), a tumor promoter and inflammatory activator, was used to stimulate proliferation and inflammatory pathways. **Methods:** HCT-116 and BJ cells were treated with SsnB (3.125–50 μM) or PMA (1–10 nM) for 12–18 h. Cell viability was assessed using MTT analysis, while apoptosis was evaluated through cleaved caspase-3 staining, terminal deoxynucleotidyl transferase dUTP nick end labeling (TUNEL), and flow cytometry. Proliferation was analyzed through proliferating cell nuclear antigen (PCNA) staining. Toll-like receptor (TLR) signaling, cytokine expression, and sphingolipid levels were measured using immunofluorescence, enzyme-linked immunosorbent assay (ELISA), and mass spectrometry, respectively. **Results:** SsnB reduced HCT-116 cell viability in a dose- and time-dependent manner with minimal effects on BJ cells. SsnB (25 μM, 12 h) decreased HCT-116 viability 0.6-fold, while PMA (10 nM, 12 h) increased it 2-fold (*p* < 0.01). No significant change was observed in BJ cells. PCNA fluorescence staining increased 2-fold with PMA and decreased 0.4-fold with SsnB (*p* < 0.001). PMA upregulated TLR2 and TLR4 mRNA and protein levels, with MyD88, p-ERK, and pNF-κB fluorescence increasing 2.1-, 1.5-, and 1.7-fold, respectively (*p* < 0.001). PMA elevated TNF-α, IL-1β, and IL-6 levels (*p* < 0.01). SsnB suppressed PMA-induced effects and promoted apoptosis, increasing cleaved caspase-3 levels by 1.5-fold and TUNEL staining by 1.9-fold (*p* < 0.01). Flow cytometry confirmed a significant increase in early and late apoptotic cells in the SsnB group. SsnB also increased ceramide (C18, C20, C22, and C24) levels (1.3- to 2.5-fold, *p* < 0.01) while reducing PMA-induced S1P and C1P increases (*p* < 0.01). **Conclusions:** SsnB selectively inhibits proliferation, induces apoptosis, and modulates inflammatory and sphingolipid pathways in colorectal cancer cells, with minimal toxicity to healthy fibroblasts, supporting its potential as a targeted therapeutic agent.

## 1. Introduction

Colon cancer is one of the most important oncological problems in developed countries and ranks second among all cancer types in terms of incidence worldwide [1]. According to data from the World Health Organization (WHO), in 2020, more than 1.9 million people worldwide were diagnosed with colon cancer and approximately 1 million people died due to colon cancer [1]. Colon cancer, which represents one out of every 10 cancer cases and deaths, has higher incidence and mortality rates in men than in women [1]. Combination therapy of colon cancers with polyphenols and chemotherapy has been one of the strategies used in cancer treatment in recent years [2,3]. Increasing the effectiveness of certain cancer drugs reduces the cost of treatment by shortening the length of hospital stay for patients and provides social impact. Therefore, it is considered necessary to develop this approach. SsnB has been classified as a polyphenol isolated from the plant roots of *Sparganium stoloniferum* [4]. Its structure is like classes of polyphenolic compounds such as isocoumarins and xanthones [5]. The structure is shown below in Figure 1 [6] (data are computed by PubChem, https://pubchem.ncbi.nlm.nih.gov, accessed on 28 January 2025).

In this study, the efficacy of SsnB, a polyphenol that has not yet been studied as a TLR 2 and TLR 4 antagonist in colon cancer, was investigated. Toll-Like receptors recognize pathogen-associated molecular patterns (PAMPs) and regulate inflammatory responses as important components of the innate immune system [7]. However, chronic activation of TLRs leads to the formation and progression of the tumor microenvironment in many types of cancer, especially colon [8]. It has been reported that overexpression of TLR 4 promotes tumor growth and metastasis in colon cancer due to inflammation [9]. In this context, the inhibition of TLR signaling pathways is considered as a promising approach in the treatment of colon cancer. Studies have reported that TLR 2 and 4 are expressed in the human HCT-116 colon cancer cell line [10,11,12]. The activation of TLR 2 and 4 signal transduction pathways in the human colon cancer cell line induces ERK phosphorylation, inducing cell proliferation via NF-kB and suppressing apoptosis [13]. SsnB has been shown to reduce NF-kB expression by suppressing TLR 2 and TLR 4 in gastric cancer cells and adipocytes [14,15,16]. Based on these findings, we hypothesized that SsnB could be antiproliferative in colon cancer cells through TLR 2 and TLR 4 signaling pathways. To the best of our knowledge, the antiproliferative effect of SsnB in colon cancer has not been investigated through the TLR 2 and TLR 4 signaling pathways.

The effects of sphingolipid metabolism on cancer development and progression have gained importance in recent years [17]. In a recently published study by our group, we discovered that administering 25 µM SsnB to estrogen receptor-positive breast and ovarian cancer cells dramatically reduced their S1P levels and caused a substantial buildup in intracellular concentrations of ceramides [18]. The formation of physiologically active sphingolipids is now understood to have an important role in the genesis and progression of cancer. Ceramide plays a crucial role in the metabolism of sphingolipids and suppresses tumor growth in a variety of cancerous cells by triggering apoptosis and anti-proliferative reactions [19]. According to certain studies, there may also be a connection between TLR signaling and the metabolism of sphingolipids, particularly the synthesis of ceramide, which is involved in cancer treatment [20]. Therefore, the effect of SsnB on ceramide, S1P, and C1P levels was also determined in human colon cancer cells.

## 2. Results

### 2.1. Effect of Sparstolonin B on Cell Viability

The effect of SsnB (3.125–50 μM) on HCT-116 and BJ cell viability at 12, 16, and 18 h is shown in Figure 1. Administration of 25 and 50 μM SsnB to HCT-116 cells significantly reduced cell viability at all hours relative to the control, DMSO, and other dose groups (*p* < 0.01) (Figure 1A). In cells administered 6.25, 12.5, and 25 μM SsnB, 18 h of administration within the same dose interval decreased cell viability compared to other hours (*p* < 0.01). In healthy human fibroblast cells (BJ), 50 μM SsnB significantly reduced cell viability at all time intervals compared to the control, DMSO, and other dose groups (*p* < 0.05) (Figure 1D). After 12 h SsnB incubation, the viability of HCT-116 cells was calculated as 58.49 ± 9.34% (mean ± SD), indicating approximately 50% cell loss. In BJ cells, 12 h 25 μM SsnB incubation did not significantly affect cell viability. Thus, 12 h incubation of 25 μM SsnB was the selected dose for other experiments. PMA was used to increase proliferation in HCT-116 cells and applied to the positive control group. Its effect on cell viability was investigated in the 12–18 h incubation interval. As shown in Figure 1B, 12-h incubation of PMA in the range of 1–10 nM effectively stimulated proliferation in HCT-116 cells (*p* < 0.01) and increased viability compared to other hours in the same dose groups (*p* < 0.01). At the end of twelve-hour PMA incubation, viability in HCT-116 cells was calculated as 195.04% ± 44.94 (mean ± SD), indicating an approximately 100% cell increase. There was no effect of PMA on increasing cell viability in BJ cells (Figure 1E), and 10 nM PMA incubation at 16 and 18 h significantly reduced cell viability compared to the other groups (*p* < 0.05). As a result of the data obtained, PMA was applied at a dose of 10 nM for 12 h in all other experiments.

In Figure 1C and Figure 1F, HCT-116 and BJ cells were incubated with 1 μL/mL DMSO, 25 μM SsnB, and/or 10 nM PMA for 12 h, respectively. In HCT-116 cells, incubation with PMA significantly increased cell viability compared to all other groups (*p* < 0.05), while incubation with SsnB significantly decreased cell viability compared to all other groups (*p* < 0.05). Cell viability was significantly reduced in the PMA + SsnB group (*p* < 0.05) and it was observed that SsnB suppressed PMA activity. In healthy human fibroblast cells, the same dose groups did not produce a significant change in cell viability (Figure 1F). Light microscope images of HCT-116 and BJ cells are given in Figure 1G. While no morphological change was observed in HCT-116 cells in the control and DMSO groups, significant proliferation was observed in the PMA (10 nM, 12 h) group. Twelve-hour administration of 25 μM SsnB altered the morphology of HCT-116 cells, resulting in clustering, shrinkage, and toxicity. No significant morphological changes were observed in BJ cells.

Figure 1H shows PCNA immunofluorescence staining in HCT-116 and BJ cells, while Figure 1I and Figure 1K report the amount of PCNA fluorescence staining, respectively. PCNA immunofluorescence staining and quantitation gave similar results in BJ cells in all experimental groups. In HCT-116 cells, after 12 h of incubation, PMA (10 nM) application brought PCNA fluorescence staining to 213.34% ± 38.04 (mean ± SD) and showed an increase of approximately 100% compared to the control. After twelve hours of incubation, SsnB (25 μM) application reduced PCNA fluorescence staining in HCT-116 cells to 37.51 ± 5.89 (mean ± SD), indicating a reduction of approximately 40% compared to the control. The amount of PCNA fluorescence in the SsnB group did not show a significant change compared to the PMA + SsnB group. The PCNA quantitation data obtained in HCT-116 cells are consistent with cell viability results. The amount of PCNA protein was also measured using the ELISA method in HCT-116 and BJ cells, and the results are reported in Figure 1J and Figure 1L, respectively.

The measured PCNA protein levels generally corresponded with the amount of fluorescence staining. However, the amount of PCNA fluorescence staining did not differ significantly between the SsnB and PMA + SsnB groups in HCT-116 cells, while protein levels increased significantly in the PMA + SsnB group compared to the SsnB group (*p* < 0.01).

### 2.2. Effect of Sparstolonin B on the TLR2-TLR4 Signaling Pathway

Figure 2A and Figure 2B show the amount of TLR2 and TLR4 mRNA, respectively. Figure 2C and Figure 2D report the amount of TLR2 and TLR4 protein, respectively. TLR2 mRNA levels showed a statistically significant increase (*p* < 0.05) in cells treated with PMA (10 nM) and PMA (10 nM) + SsnB (25 μM) for 12 h compared to the group of cells treated with SsnB (25 μM) alone (Figure 2A). In accordance with the mRNA amounts, TLR2 protein levels were also significantly increased in the groups treated with PMA and PMA + SsnB compared to the other groups (*p* < 0.001) (Figure 2C). TLR4 mRNA levels showed a statistical increase in the PMA-treated cell group compared to the SsnB group (*p* < 0.01) (Figure 2B). In accordance with mRNA levels, the amount of TLR4 protein was significantly higher in cells that received PMA compared to all other groups (*p* < 0.001) (Figure 2D).

Figure 2E shows MyD88, p-ERK, and p-NF-kB immunofluorescence staining. Figure 2F and Figure 2G report the quantitation of MyD88 fluorescence staining and ELISA protein measurement results, respectively. MyD88 fluorescence staining and protein levels were significantly increased in PMA-treated cells compared to all other groups (*p* < 0.001). While MyD88 fluorescence staining was significantly reduced in cells incubated with SsnB compared to the control and DMSO groups (*p* < 0.001), no statistically significant decrease in the amount of protein was detected. Similarly, while the amount of MyD88 fluorescence staining in the PMA + SsnB group did not show a statistically significant difference compared to the control and DMSO groups, ELISA protein measurement results indicated a statistically significant increase compared to the other groups (*p* < 0.001).

Figure 2H,I report the quantitation results of fluorescence staining of p-ERK and p-NF-kB, respectively. While p-ERK and p-NF-kB fluorescence staining increased significantly in PMA cells compared to all other groups (*p* < 0.01), p-ERK and p-NF-kB fluorescence staining decreased significantly in cells incubated with SsnB compared to all other groups (*p* < 0.001).

Figure 2J, Figure 2K, and Figure 2L show the protein levels of TNF-α, IL-1B, and IL-6, respectively. TNF-α cytokine levels were significantly increased in the PMA group compared to the control, DMSO, and SsnB groups (*p* < 0.05) (Figure 2J). IL-1B and IL-6 levels increased significantly in the PMA group compared to all other groups (*p* < 0.05) and decreased significantly in the SsnB group (*p* < 0.001).

### 2.3. Effect of Sparstolonin B on Apoptosis

Figure 3A shows cleaved caspase-3 and TUNEL fluorescence staining. Figure 3B,C report the quantity of cleaved caspase-3 and TUNEL fluorescence staining, respectively. The amount of cleaved caspase-3 fluorescence staining was significantly increased in the SsnB-treated group compared to the control, DMSO, and PMA groups (*p* < 0.01). TUNEL staining was significantly increased in the SsnB-treated group compared to all other groups (*p* < 0.01). PMA application reduced TUNEL fluorescence staining, and the number of TUNEL-stained cells increased significantly in the PMA + SsnB group compared to the control group only (*p* < 0.05).

Figure 3D shows flow cytometry graphs of Annexin-FITC and PI-labeled cells. Figure 3E reports the quantitative analysis of Annexin-V- and PI-labeled cells. The number of viable cells decreased statistically in the SsnB group compared to all other groups (*p* < 0.05). The number of early apoptotic cells statistically increased in the SsnB group compared to all other groups (*p* < 0.05). The number of late apoptotic cells increased significantly in the SsnB group compared to the control and PMA groups (*p* < 0.05).

### 2.4. The Effect of Sparstolonin B on Sphingolipid Levels

Table 1 reports the levels of sphingolipids measured in the cell groups. A statistically significant increase in C18, C20, C22, and C24 ceramide levels was detected in cells treated with SsnB compared to all other groups (*p* < 0.01). A significant increase in the amount of S1P and C1P was observed in the cell groups treated with PMA compared to the other groups (*p* < 0.01). S1P levels showed a statistical decrease in the PMA + SsnB group compared to the PMA group (*p* < 0.01), but the measured levels were significantly higher than in the control, DMSO, and SsnB groups (*p* < 0.01). Similarly, C1P levels showed a statistical decrease in the PMA + SsnB group compared with the PMA group (*p* < 0.01), but the measured levels were significantly higher than the DMSO and SsnB groups (*p* < 0.05).

## 3. Discussion

The results of this study show that treating HCT-116 cells with 10 nM PMA for 12 h significantly increased cell proliferation. There was no significant change in cell proliferation in BJ fibroblast cells that received PMA at the same dose and time. PMA is a protein kinase C (PKC) activator and regulates cell proliferation and biological processes such as apoptosis through PKC activation [21,22,23]. We selected PMA as the tumor promoter and inflammatory activator in our study due to its well-documented role in stimulating proliferation, inflammation, and TLR signaling pathways, particularly in colorectal cancer models. PMA effectively mimics the tumor-promoting effects of chronic inflammation, which is a key driver in colorectal cancer progression [24]. PMA has been shown to induce the expression of Toll-like receptors (TLRs), particularly TLR2 and TLR4, in various cell types. For instance, a study demonstrated that PMA-mediated differentiation of HL-60 cells into macrophage-like cells resulted in a significant increase in TLR2 expression and, to a lesser extent, TLR4 expression. This upregulation was found to be dependent on protein kinase C activation [25]. PMA is frequently used in cancer studies, including colorectal cancer models, to study pro-inflammatory and pro-tumorigenic signaling cascades [26]. Indeed, incubation of HCT-116 cells with 25 μM SsnB for 12 h significantly reduced cell proliferation in the presence and absence of PMA. No statistically significant change in cell proliferation occurred in the presence and absence of PMA in BJ fibroblast cells treated with SsnB at the same dose and for the same period. Studies have shown that the administration of SsnB to cancer and endothelial cells reduces cell proliferation [27,28,29,30,31] SsnB has been reported to inhibit cancer-related processes such as cell migration, invasion, and angiogenesis by stopping the cell cycle at G1 or G2/M checkpoints [29]. Administration of SsnB at concentrations between 10–100 μM has been found to show potent antiproliferative effects in various cell types, including neuroblastoma [4], prostate cancer [27,31], and pancreatic cancer [30] models. Within our literature review, we found that this study is the first to examine the PMA-dependent proliferation of SsnB in HCT-116 human colon cancer and healthy BJ fibroblast cells.

A significant increase in the intracellular levels of C18, C20, C22, and C24 ceramides occurred in HCT-116 cells treated for 12 h with a concentration of 25 μM of SsnB compared to all experimental groups. This is the first study to evaluate the impact of SsnB on endogenous sphingolipid levels in human colon cancer cells. Most of the research on SsnB concentrates on its anti-proliferative, anti-inflammatory, and anti-angiogenic properties, which are mainly achieved via inhibiting NF-κB and STAT1 signaling [32] and modifying Toll-like receptor (TLR) pathways [15]. When TLR4 or TLR1/TLR2 or TLR2/TLR6 ligands were used to stimulate mouse macrophages, SsnB significantly reduced the production of inflammatory cytokines [15]. The evaluated research has not provided sufficient evidence on the effects of SsnB on lipid metabolism or ceramide levels. Ceramide production in cancer cells and TLR signaling have a complicated interaction. According to certain studies, there may be a connection between TLR signaling and the metabolism of sphingolipids, particularly the synthesis of ceramide, which is involved in cancer treatment [33]. There are not many solid examples of TLR antagonists directly boosting ceramide synthesis though. Increased TLR expression has been linked to oncogenesis and the progression of cancer in cancer cell lines [34]. TLRs regulate inflammatory responses in cancer cells. However, there is ongoing debate over the connection between TLRs and cancer. TLRs have been demonstrated to both accelerate and slow the progression of cancer [35,36,37].

Toll-like receptors have emerged as critical players in cancer biology, making them appeal as therapeutic targets due to their elevated expression in various tumors, including human colon cancer [8]. TLR2, TLR3, and TLR4 are overexpressed in most colon cancer cells [38]. However, in the context of colon cancer, aberrant TLR signaling has been linked to both tumor-promoting and tumor-suppressing effects, underscoring the complexity of targeting these pathways for therapeutic purposes [39]. For more than thirty years, bladder cancer has been treated with Bacillus Calmette–Guérin (BCG), the TLR2/TLR4 agonist that is the most effective TLR ligand in cancer therapy [40]. Its effectiveness highlights the therapeutic potential of TLR pathway modulation in oncology. During tumor development, TLR4 activation promotes the production of proinflammatory chemokines and immunosuppressive cytokines in an unregulated manner. This environment facilitates immune evasion, angiogenesis, and metastasis, as observed in human colon cancer [41]. These findings highlight the potential of TLR4 antagonists as a novel therapeutic strategy. Our findings suggest that manipulating TLR signaling holds significant therapeutic promise, particularly through the modulation of ceramide synthesis. Ceramide, a bioactive lipid, plays a crucial role in inducing cancer cell death and overcoming drug resistance [42]. In the context of TLR signaling, the small molecule SsnB has been hypothesized to influence ceramide synthesis, thereby enhancing ceramide-driven apoptosis in colon cancer cells. This mechanism involves TLR inhibition to create a pro-apoptotic environment characterized by ceramide accumulation, which, coupled with attenuated survival signaling, selectively induces apoptosis in cancer cells. Such dual modulation underscores the potential of SsnB as a targeted therapeutic agent in colon cancer characterized by dysregulated TLR pathways and survival mechanisms.

We discovered that administering 25 µM SsnB to colon cancer cells for 12 h dramatically reduced their S1P and C1P levels. As far as we are aware, our study is the first to show that colon cancer cells treated with SsnB had lower S1P levels. Examples of functional sphingolipid metabolites that are vital to the biological pathways that are fundamental to the pathophysiology of cancer are C1P and S1P [43]. Both S1P and C1P sets off mechanisms that transform it into a lipid that promotes cancer [44].

We observed that MyD88, p-ERK, p-NF-kB, TNF-α, IL-1β, and IL-6 were significantly suppressed in cancer cells treated with SsnB compared to the control groups. These findings align with previous studies indicating that SsnB inhibits key inflammatory and survival pathways in cancer cells [18]. The MyD88/NF-kB signaling axis is a central pathway in regulating inflammatory responses and tumor progression in colorectal cancer [45]. Activation of MyD88, an adaptor protein, triggers a signaling cascade involving p-ERK and NF-kB, leading to the production of pro-inflammatory cytokines such as TNF-α, IL-1β, and IL-6 [46]. These cytokines not only promote a pro-tumorigenic microenvironment but also enhance cancer cell proliferation, survival, and metastasis [46]. The suppression of MyD88 and downstream mediators like p-ERK and p-NF-kB by SsnB suggests a potent anti-inflammatory effect, which could disrupt the tumor-supportive inflammatory milieu. Notably, NF-kB is often constitutively activated in various cancers, contributing to the expression of genes involved in cell survival, invasion, and chemoresistance [47]. By inhibiting NF-kB activation, SsnB may diminish the expression of its target cytokines, including TNF-α, IL-1β, and IL-6, which are critical for maintaining cancer cell viability and immune evasion.

Our results also highlight the therapeutic potential of SsnB in targeting cytokine-mediated pathways. TNF-α and IL-1β, key pro-inflammatory cytokines, are known to enhance angiogenesis, tumor invasion, and resistance to apoptosis. IL-6 further supports tumor growth through the activation of STAT3, a transcription factor implicated in cancer cell proliferation and metastasis in colorectal cancer cells [48]. The downregulation of these cytokines by SsnB may mitigate their tumor-promoting effects and enhance cancer cells’ susceptibility to apoptosis.

In summary, SsnB exerts its anti-cancer effects by downregulating MyD88, p-ERK, and p-NFκB signaling, along with suppressing the pro-inflammatory cytokines TNF-α, IL-1β, and IL-6. This comprehensive inhibition disrupts the tumor-supportive microenvironment and enhances apoptosis, offering a promising therapeutic approach for cancer treatment. This study’s limitations include its reliance on HCT-116 cancer cells and BJ fibroblasts, which do not fully represent cancer heterogeneity or normal cell behavior. The in vitro model cannot replicate complex in vivo tumor dynamics, including immune and stromal interactions. PMA, used to induce inflammation, may not mimic natural tumor promotion. The limited dose range and treatment duration for SsnB leave long-term efficacy and safety unexplored.

Prospective studies should explore the effects of SsnB in additional colorectal cancer and non-cancerous epithelial cell lines to confirm broader applicability. Further mechanistic investigations are needed to elucidate its role in apoptosis and inflammatory signaling pathways. In vivo studies will be crucial to assess its therapeutic efficacy, safety, and potential impact on the tumor microenvironment. Additionally, evaluating SsnB in combination with chemotherapeutic agents may provide insights into its potential to enhance treatment efficacy or overcome drug resistance. Finally, understanding its role in sphingolipid metabolism may uncover novel therapeutic targets. These future directions will help establish SsnB’s potential as a candidate for colorectal cancer treatment.

## 4. Materials and Methods

### 4.1. Materials

In this study, we utilized high-purity chemicals from various suppliers. Dulbecco’s Modified Eagle’s Medium (DMEM, Sigma-Aldrich, Cat. #D5648) was supplemented with sodium bicarbonate (Sigma-Aldrich), fetal bovine serum (FBS, Gibco), penicillin (Gibco), streptomycin (Gibco), sodium pyruvate (Gibco), and amphotericin B (Gibco). Sparstolonin B (SsnB, ≥98%) was purchased from Merck Millipore (Cat. #SML1767) and dissolved in dimethyl sulfoxide (DMSO, Sigma-Aldrich, ≥99.9%). Phorbol 12-Myristate 13-Acetate (PMA, ≥98%) was obtained from LC Laboratories (Cat. #P-1680) and dissolved in DMSO (Sigma-Aldrich, ≥99.9%). MTT (≥99%) was from Gold Biotechnology and dissolved in phosphate-buffered saline (PBS, Sigma-Aldrich). Immunofluorescence reagents included paraformaldehyde, Triton X-100 (Sigma-Aldrich), normal goat serum (Vector Laboratories), and primary antibodies from Bioss Antibodies, Affinity Biosciences, BT Lab, and Cell Signaling Tech., with secondary antibody Alexa Fluor (Abcam) and DAPI (Sigma-Aldrich). The ELISA kits were purchased from BT Lab and Biotechnology. The TUNEL assay kit was purchased from Elabscience. Flow cytometry reagents included Annexin-V and propidium iodide (Elabscience). Sphingolipidomic reagents included chloroform, methanol, ammonium formate, and formic acid (Sigma-Aldrich). Standards were purchased from Avanti Polar Lipids, and the internal standard was purchased from Cambridge Isotope Laboratories. LC-MS/MS analysis involved the use of the LCMS-8040 system (Shimadzu) with an XTerra C18 column (Waters).

### 4.2. Cell Culture

Human colon cancer (HCT-116) and healthy human fibroblast cell lines (BJ) were obtained from the American Type Culture Collection (ATCC; Manassas, VA, USA). Cells were cultured in 25 cm2 culture flasks using Dulbecco’s Modified Eagle’s Medium (DMEM, Sigma, Cat. #D5648, St. Louis, MO, USA) with high glucose. The medium was supplemented with, 3.7 g/L sodium bicarbonate (Sigma-Adrich), 10% (*v*/*v*) heat-inactivated fetal bovine serum (FBS) (Gibco, Life Technologies, Grand Island, NY, USA), 100 U/mL penicillin (Gibco), 100 μg/mL streptomycin (Gibco), 1% sodium pyruvate (Gibco), and 5 μg/100 mL amphotericin B (Gibco). The prepared medium was sterilized with vacuum filtering through a 0.22 μm bottle-top filter and stored at 4 °C. The cells were incubated at 37 °C with 5% CO_2_ and 95% humidified air. When the cells reached 80% density, they were lifted from the flask surface with trypsin-EDTA (0.05% Trypsin/0.02% EDTA; Gibco) and suspended and passed to new flasks.

### 4.3. Application of Sparstolonin B and Phorbol 12-Myristate 13-Acetate

Five mg SsnB (MW = 268.22 g/mol; Merck Millipore, Cork, Ireland, #SML1767) was dissolved in 1.854 mL of DMSO to yield a stock concentration of 10 mM. The prepared main stock was diluted with DMSO to create an intermediate stock of 1 mM. One mM SsnB was diluted with cell culture medium to concentrations of 3.125, 6.25, 12.5, 25, and 50 μM. MTT [3-(4,5-dimethylthiazol-2-yl)-2,5-diphenyltetrazolium bromide] cell viability analysis was used to determine the dose and duration of application.

Next, 5 mg PMA (MW = 616.83 g/mol; LC Laboratories, Canada, #P-1680) was dissolved in 1 mL DMSO to prepare a stock concentration of 8.1 mM. The prepared main stock was diluted with DMSO to create an intermediate stock of 1 mM. Thereafter, 1 mM of PMA intermediate stock was diluted with cell culture medium to concentrations of 1, 5, and 10 nM. The dose and duration of administration were determined through MTT analysis.

### 4.4. Cell Viability Analysis

MTT (Gold Biotechnology Inc., St. Louis, MO, USA) was dissolved at a concentration of 5 mg/mL in PBS and sterilized via passage through a filter with a pore diameter of 0.22 μm. Before MTT analysis, HCT-116 and BJ fibroblast were plated in 96 well plates and left for one night to adhere. After adhesion, 200 μL of cell culture medium containing either DMSO (1 μL/mL), PMA (1–10 nM) and/or SsnB (3.125–50 μM) were added to the wells. After 12–18 h of incubation, the media were withdrawn, fresh medium containing MTT was added (90 μL medium + 10 μL MTT with a total volume of 100 μL) and the 96 well plate was incubated for 2 h. Purple formazan crystals formed at the bottom of the wells were dissolved with 100 μL of DMSO and absorbance was measured at 570 and 690 nm using a spectrophotometric plate reader (MicroQuant, Bio-Tek Instruments Inc., Charlotte, VT, USA). The percentage of cell viability was calculated so that the absorbance in the control group represented 100% cell viability [Cell viability (%) = (Abs sample/Abs control) × 100]. According to the results of the MTT test, five different experimental groups were formed for each cell line.

### 4.5. Immunofluorescence Staining

Approximately 100,000 cells/well were plated one an 8-well chamber slide (Merck Millipore, Cork, Ireland) and the cells were allowed to adhere overnight. Following 12 h incubation of the treatment groups, the cells were washed twice with cold PBS and were fixed in 4% paraformaldehyde (Sigma-Aldrich, St. Louis, MO, USA) solution for 10 min. After two washes in PBS, the cells were permeabilized in 0.2% Triton-X-100 (Sigma-Aldrich, St. Louis, MO, USA) for 10 min and washed five times with PBS. Afterward, 5% normal goat serum (NGS; Vector Laboratories, Burlingame, CA, USA) was used for blocking for 30 min. Next, 200 μL of the primary antibody was applied, and the chamber slide was kept at 4 °C overnight. Primary antibodies were diluted with Ab diluent (PBS containing 5% NGS and 0.2% Tween 20). The primary antibodies used were PCNA rabbit polyclonal ab (1:100 dilution, #bs-0754R, Bioss Antibodies Inc., Woburn, MA, USA), phospho-ERK1/2 (Thr202/Tyr204) rabbit polyclonal ab (1:200 dilution, #AF1015, Affinity Biosciences, Changzhou, Jiangsu, China), phospho-NF-kB rabbit polyclonal ab (1:200 dilution, #AF2006, Affinity Biosciences, Changzhou, Jiangsu, China), MyD88 rabbit polyclonal ab (1:100 dilution, # BT-AP05682, BT Lab, Shanghai, China), and cleaved-caspase 3 rabbit polyclonal ab (1:100 dilution, #9664S, Cell Signaling Tech., Danvers, MA, USA). After primary antibody incubation, the incubation medium was withdrawn, three washes were performed with PBS, and 200 μL of fluorophore-labeled secondary antibody (Alexa Fluor, 1:1000 dilution, ab150077 Abcam, Cambridge, UK) was added. The cells were incubated at room temperature for 45 min, and after incubation, the cells were washed with PBS twice and a drop of DAPI was applied (Fluoroshield with DAPI #F6057, Sigma Aldrich, St. Louis, MO, USA) to stain the cell nuclei. The slides were covered with coverslips without air bubbles. Imaging was performed with a fluorescence microscope (Olympus BX61 fully automated, Tokyo, Japan) using DP controller 3.2.1.276 software. Alexa Fluor was imaged at 488 nm excitation and 505–525 nm emission, while DAPI was imaged at 350 nm excitation and 440–460 nm emission. Fluorescence was measured using NIH ImageJ 1.53e software and calculated with the formula CTCF (corrected total cell fluorescence) = ID (integrated density) − [ASC (area of selected cell) × BMF (background mean fluorescence)].

### 4.6. Determination of TLR 2 and TLR 4 mRNA Expression

The OMIM (Online Mendelian Inheritance in Man) database was used as a source for all mRNA sequences. OligoYap 9.0 software (SNP Biotechnology R&D Ltd., Ankara, Turkey) was used to design the primers and probes. Appendix A lists the primers and probes that were used. The One-run RT PCR kit (SNP Biotechnology R&D Ltd., Ankara, Turkey) was used to conduct real-time PCR analysis. Several forward and reverse primer concentrations were used in the optimization process to determine the lowest primer concentration that would produce the highest ΔRn (the difference between the sample’s minimum and maximum fluorescence). Using various probe concentrations at the ideal primer concentrations allowed for the determination of the ideal probe concentration. The concentration of the probe that yielded the lowest cycle threshold (Ct) value was found to be the optimal one.

Total RNA extraction was performed using a commercial kit and the manufacturer’s instructions (SNP Biotechnology R&D Ltd., Ankara, Turkey). The extracted RNA was dissolved in 70 μL of TE buffer [10 mM Tris-HCl, 0.1 mM EDTA (pH 7.5)]. Next, 10 μL of the dissolved RNA was diluted with 490 μL of distilled water (1:50 dilution). The diluted RNA sample was tested for absorbance at 260 and 280 nm using spectrophotometry. The purity of the RNA samples was assessed using the test results at 260 and 280 nm. Samples with a 260 nm/280 nm ratio of less than two were suitable for analysis. The absorbance at 260 nm was used to calculate the samples’ RNA content. At 260 nm, 40 μg/mL of RNA was equivalent to an absorbance unit of 1. Sample RNA (100 ng/μL), one-run mix, primer-1 (100 pmol), primer-2 (100 pmol), and the probe (100 pmol) were all included in the sample preparation for each assay.

RT-PCR analysis was performed using the Mx3000p Multiplex Quantitative PCR system (Stratagene, San Diego, CA, USA). The cycle conditions were as follows: step 1, 10 min at 42 °C (1 cycle); step 2, 5 s at 90 °C; and step 3, 45 s at 60 °C (40 cycles). Using MxPro QPCR v6.22 Software for the Mx3000P system, the log-linear phase of amplification was monitored to obtain Ct values for each RNA sample. All reactions were conducted in three replicates, and the mRNA levels of each gene normalized to beta-actin were expressed as a 2^−(ΔΔct)^ fold change.

### 4.7. ELISA Measurements

PCNA was determined with a non-competitive sandwich ELISA kit (cat. #ELK5141 Biotechnology; Denver, CO, USA). HCT-116 and BJ fibroblasts (10^7^ cells/mL) were subjected to ultrasonication, and supernatants were collected through centrifugation at 4 °C for 10 min at 1500× *g*. The amount of PCNA was determined in accordance with the kit instructions and the samples were measured with a spectrophotometric plate reader at 450 nm. The PCNA concentration in the samples was calculated using a standard curve and reported as ng/cell count.

Protein levels of TLR2 (cat # E0358Hu. BT Lab, Shanghai, China), TLR4 (cat # E0346Hu. BT Lab, Shanghai, China), MyD88 (cat. #E1870Hu BT Lab, Shanghai, China), TNF-α (cat. #E0082Hu. BT Lab, Shanghai, China), IL-1β (cat # E0143Hu BT Lab, Shanghai, China), and IL-6 (cat. #E0090Hu BT Lab, Shanghai, China) were assessed using sandwich ELISA kits in cell lysates. Cell pellets from the experimental groups were suspended in 250 μL of cold PBS for the cell samples. Sonication (Bandelin Sonopuls HD 2070, Bandelin Elec., Berlin, Germany) completely broke up the cells. After centrifuging the lysates at 1500× *g* for 10 min at 2–8 °C, the supernatants were gathered. Spectrophotometric absorbance measurements were taken at 450 nm using the measurement procedures outlined in the kit manuals. To analyze sample concentrations on nonlinear standard curves, GraphPad Prism software was used for curve fitting and interpolation. The results were expressed in milligrams of cell protein.

### 4.8. TUNEL Analysis

Cell death was detected using the One-Step TUNEL Assay Kit (Elabscience; Houston, TX, USA, E-CK-A320). This technique relies on the use of fluorescent markers to identify double-stranded DNA breaks that take place in apoptotic cells. The test protocol was followed in the preparation of each sample and reagent. On the chamber slides (Merck Millipore, Cork, Ireland), 100,000 cells were transferred per well. After treatment and incubation according to the experimental groups, the cells were fixed using fixative buffer (paraformaldehyde) for 20 min at room temperature. The slides were washed three times with PBS for five minutes each after fixation. DNase I (200 U/mL) was added to create positive controls, which were then incubated for 25 min at 37 °C. Negative controls did not receive TdT Enzyme treatment; instead, they were incubated with an identical volume of buffer. All other groups received 100 μL of TdT, which was then incubated for 25 min at 37 °C. Following incubation, each slide received 50 μL of labeling working solution, which was then incubated for 60 min at 37 °C in a humidified environment. The slides then went through three 5 min PBS washes. Upon the removal of the chambers and a tissue-cleaning procedure, a drop of DAPI (Sigma-Aldrich; St. Louis, MO, USA) was applied to each slide. A fluorescence microscope (Olympus BX61 fully automated, Tokyo, Japan) was used to view the fluorescence intensity after a clean, air-bubble-free coverslip was put in place.

### 4.9. Determination of Apoptotic Cells via Flow Cytometry

The apoptotic effects of PMA and SsnB on HCT-116 and BJ cells were assessed using an Annexin-V/PI apoptosis kit (Elabscience: #E-CK-A211, Houston, TX, USA). The cells were treated as previously described, washed with PBS, and detached using Trypsin-EDTA. After suspension in PBS, 1 × 10^6^ cells were transferred to flow cytometry tubes, washed twice, and centrifuged at 125× *g* for 5 min. The supernatant was discarded, and cells were resuspended in 500 µL Annexin-V binding buffer. Annexin-V-FITC (5 µL) and PI (5 µL) were added, and the mixture was incubated for 20 min at room temperature. Fluorescently labeled cells were analyzed immediately on the FACS Canto II flow cytometer (BD Biosciences, San Jose, CA, USA) with appropriate settings. Data were processed using BD FACS Diva 6.1.3 software, and the results were expressed as percentages of total cell staining.

### 4.10. Sphingolipidomic Analysis

The samples were prepared for SM and CER analysis as previously described [49]. Cell lysates (100 mg/mL protein) were mixed with 2 μL of a 5000 ng/mL internal stock solution, and then 375 μL of chloroform–methanol (1:2, *v*/*v*) was added. After 30 s of sonication and 5 min of vortexing with 100 μL of distilled water, the mixture was allowed to rest at room temperature for 30 min. The mixture was centrifuged at 2000× *g* for five minutes after incubation, and the supernatant was gathered. After adding 125 μL of distilled water and 125 μL of chloroform, the supernatant was vortexed and allowed to sit at room temperature for an additional half hour. After incubation, a nitrogen stream (VLM, Bielefeld, Germany) was used to dry 500 μL of the top organic layer in a new glass tube. The dried residue was dissolved in 100 μL of methanol–formic acid (99.9:0.1) and then placed into insert vials for LC-MS/MS analysis.

Ceramide and sphingomyelin levels were measured using an LC/MSMS instrument (LCMS-8040, Shimadzu Corporation, Kyoto, Japan) paired with an ultra-fast liquid chromatography system (LC-20 AD UFLC XR) [46]. Standards were obtained from Avanti Polar Lipids, with a labeled internal standard (C16 CER d18:1/16:0) from Cambridge Isotope Laboratories. Sphingolipid standards were prepared through sonication at 40 °C in methanol. Chromatographic separation was achieved on an XTerra C18 HPLC column (2.1 mm × 50 mm, Waters, Milford, MA, USA) at 60 °C with a 10 μL injection volume and a 0.450 mL/min flow rate. Gradient elution lasted 30 min with mobile phases of water/acetonitrile/2-propanol and acetonitrile/2-propanol. Positive electrospray ionization (ESI) and multiple reaction monitoring (MRM) were used for detection. Calibration ranges were linear between 39 and 625 ng/mL.

### 4.11. Protein Measurements

The protein concentration in all samples was determined at 595 nm using Coomassie reagent (Thermo Scientific; Rockford, IL, USA). Bovine Serum Albumin was used as standard in this measurement.

### 4.12. Statistical Analyses

SigmaPlot 15 or GraphPad Prism 8.4.3 were used for statistical analyses. Normality tests determined the data distribution. Non-parametric tests were used for non-normal data. Experimental groups were compared using Kruskal–Wallis or one-way ANOVA, followed by post-hoc tests to identify specific group differences when significant differences were found. The results are detailed in figure legends.

## Data Availability

Data obtained and analyzed in this work are available from the corresponding author upon reasonable request.

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
