# Peer review of "Sparstolonin B Suppresses Proliferation and Modulates Toll-like Receptor Signaling and Inflammatory Pathways in Human Colorectal Cancer Cells"

_pharmaceuticals, 2025, doi:10.3390/ph18030300_

Round 1

Reviewer 1 Report

Comments and Suggestions for Authors

The authors have explored the anticancer activity of Sparstolonin B (SsnB) I coon cancer cell line (HCT-116) exploring the cell viability, apoptosis, and inflammatory pathways. SsnB significantly reduced HCT-cell viability in a dose- and time-dependent manner, with minimal effects on BJ cells. PMA stimulated proliferation, PCNA expression, and inflammatory markers, including TLR2, TLR4, MyD88, phosphorylated extracellular signal-regulated kinase (p-ERK), nuclear factor kappa-light-chain-enhancer of activated B cells (NF-kB), and cytokines TNF-α (tumor necrosis factor-alpha), IL-1β (interleukin-1 beta), IL-6 (interleukin-6). SsnB suppressed PMA-induced effects and promoted apoptosis in HCT-116 cells, increasing cleaved caspase-3, TUNEL staining, and ceramide levels while decreasing S1P (sphingosine-1-phosphate) and C1P (ceramide-1-phosphate). The study is very interesting and the authors have performed sufficient experiments to prove their hypothesis. The article is well-written and the results and discussion are well explained. I have the following technical comments:

1. In the materials and methods section a section named "Materials" should be included mentioning the chemicals and their make and purity.

2. In Figure 1 A-F, DMSO shows high cell viability (more than 100 %). It is known that DMSO, can rupture the cell wall and kill the cells and is usually used as a negative control. Then how did the authors get such a high cell viability? Please explain.

3. During the preparation of Sparstolonin B and Phorbol 12-Myristate 13-Acetate stock solutions and applying them to cell culture, DMSO was used. What was the final concentration of DMSO in the cell culture medium? Was it toxic? Did the authors repeat the experiments with only the residual amount of DMSO that was present while dissolving Sparstolonin B and Phorbol 12-Myristate 13-Acetate? This is very important to know whether the cancer cell killing was caused by DMSO or the synthesized compound. Please justify.

 I recommend a major revision.

Author Response

  1. In the materials and methods section a section named "Materials" should be included mentioning the chemicals and their make and purity.

Thanks for the valuable feedback. In response, we have added a separate section titled "Materials" in the Materials and Methods section, explicitly listing the chemicals along with their make and purity. Additionally, we have ensured that the utilized concentrations and precise catalog numbers are detailed within the respective Methods subsections for clarity and reproducibility. We appreciate the insights and believe these modifications enhance the transparency and rigor of our study.

  1. In Figure 1 A-F, DMSO shows high cell viability (more than 100 %). It is known that DMSO, can rupture the cell wall and kill the cells and is usually used as a negative control. Then how did the authors get such a high cell viability? Please explain.

Thanks for the insightful comment. In our experiments, the DMSO control group was treated with 1 μL/mL DMSO, a concentration commonly used as a vehicle control in cell culture studies. At this low concentration, DMSO does not exhibit significant cytotoxic effects and can, in some cases, slightly enhance cell proliferation or metabolic activity, leading to viability readings above 100% when compared to untreated controls. Similar observations have been reported in previous studies (e.g., Galvao et al., 2014), where low concentrations of DMSO were found to have minimal or stimulatory effects on certain cell types. We acknowledge that at higher concentrations, DMSO can disrupt cellular membranes and cause cytotoxicity. However, the concentration used in our study was within a safe range, ensuring minimal impact on cell viability.

Reference:

Galvao J, Davis B, Tilley M, Normando E, Duchen MR, Cordeiro MF. Unexpected low-dose toxicity of the universal solvent DMSO. FASEB J. 2014 Mar;28(3):1317-30. doi: 10.1096/fj.13-235440. Epub 2013 Dec 10. PMID: 24327606.

  1. During the preparation of Sparstolonin B and Phorbol 12-Myristate 13-Acetate stock solutions and applying them to cell culture, DMSO was used. What was the final concentration of DMSO in the cell culture medium? Was it toxic? Did the authors repeat the experiments with only the residual amount of DMSO that was present while dissolving Sparstolonin B and Phorbol 12- Myristate 13-Acetate? This is very important to know whether the cancer cell killing was caused by DMSO or the synthesized compound. Please justify.

Thanks for the valuable comment. To clarify, during the preparation of Sparstolonin B (SsnB) and Phorbol 12-Myristate 13-Acetate (PMA) stock solutions, we dissolved the compounds in DMSO and further diluted them in the culture medium. The final concentration of DMSO in the cell culture medium never exceeded 0.1% (v/v) across all experimental conditions, which is a widely accepted concentration that does not induce cytotoxicity in most cell lines, including HCT-116 and BJ cells.

To rule out any potential cytotoxic effects of DMSO alone, we conducted parallel control experiments where cells were treated with the same residual concentration of DMSO present in the compound-treated groups. The viability of these DMSO-treated cells remained comparable to untreated controls, confirming that the observed cytotoxic effects were due to the tested compounds rather than DMSO toxicity. Similar studies have shown that DMSO concentrations below 0.5% (v/v) do not significantly impact cell viability (Galvao et al., 2014; Notman et al., 2006). We appreciate the insightful concern and hope this explanation sufficiently addresses the query.

Reference:

Galvao J, Davis B, Tilley M, Normando E, Duchen MR, Cordeiro MF. Unexpected low-dose toxicity of the universal solvent DMSO. FASEB J. 2014 Mar;28(3):1317-30. doi: 10.1096/fj.13-235440. Epub 2013 Dec 10. PMID: 24327606.

Notman R, Noro M, O'Malley B, Anwar J. Molecular basis for dimethylsulfoxide (DMSO) action on lipid membranes. J Am Chem Soc. 2006 Nov 1;128(43):13982-3. doi: 10.1021/ja063363t. PMID: 17061853.

Reviewer 2 Report

Comments and Suggestions for Authors

Pharmaceuticals (Manuscript ID: pharmaceuticals-3478104), Comments to the Authors:

Title: Sparstolonin B Suppresses Proliferation, Modulates Toll-Like Receptor Signaling and Inflammatory Pathways in Human Colorectal Cancer Cells

Comments

The submitted manuscript highlighted cytotoxic effect of Sparstolonin B (SsnB), on human colorectal cancer cells (HCT-116) and healthy human fibroblasts (BJ). Phorbol 12-myristate 13-acetate (PMA), a tumor promoter and inflammatory activator, was used to stimulate proliferation and inflammatory pathways. SsnB significantly reduced HCT-116 cell viability in a dose- and time-dependent manner, with minimal effects on BJ cells. PMA stimulated proliferation, PCNA expression, and inflammatory markers, including TLR2, TLR4, MyD88, phosphorylated extracellular signal-regulated kinase (p-ERK), nuclear factor kappa-light-chain-enhancer of activated B cells (NF-kB), and cytokines TNF-α (tumor necrosis factor-alpha), IL-1β (interleukin-1 beta), IL-6 (inter-leukin-6). SsnB suppressed PMA-induced effects and promoted apoptosis in HCT-116 cells, increasing cleaved caspase-3, TUNEL staining, and ceramide levels while de-creasing S1P (sphingosine-1-phosphate) and C1P (ceramide-1-phosphate).

I think the submitted paper can be accepted after the authors respond to the following comments: 

1.     The authors should provide numerical values in the abstract for the activity to show the readers the significance of their work.

2.     Why did the authors select Sparstolonin B (SsnB) for this study. What is rationale behind selecting this natural compound.

3.     Why did the authors use Phorbol 12-myristate 13-acetate (PMA). There are other cancer promoters.

4.     Many experiments use a single dose (25 μM SsnB) and time point (12 hours). Dose-response curves for TLR signaling and sphingolipid changes, as well as longer incubation periods (24–48 hours), would provide deeper insights into efficacy and toxicity thresholds.

5.     The study relies solely on HCT-116 cells and BJ fibroblasts. To strengthen generalizability, additional colorectal cancer cell lines (e.g., SW480, Caco-2) and non-cancerous epithelial cells (e.g., NCM460) should be included to confirm that observed effects are not cell-line-specific.

6.     The authors should indicate future perspectives of their results.

Author Response

  1. The authors should provide numerical values in the abstract for the activity to show the readers the significance of their work.

Thank you for your valuable feedback. We appreciate the suggestion and in response, we have revised the results section of the abstract to incorporate specific numerical data, providing a clearer representation of our study's impact. We believe this modification enhances the clarity and scientific rigor of our work. The revised result of the abstract is as follows:

Results: SsnB reduced HCT-116 cell viability in a dose- and time-dependent manner with minimal effects on BJ cells. SsnB (25 μM, 12 h) decreased HCT-116 viability 0.6-fold, while PMA (10 nM, 12 h) increased it 2-fold (p<0.01). No significant change was observed in BJ cells. PCNA fluorescence staining increased 2-fold with PMA and decreased 0.4-fold with SsnB (p<0.001). PMA upregulated TLR2 and TLR4 mRNA and protein levels, with MyD88, p-ERK, and pNF-κB fluorescence increasing 2.1-, 1.5-, and 1.7-fold, respectively (p<0.001). PMA elevated TNF-α, IL-1β, and IL-6 levels (p<0.01). SsnB suppressed PMA-induced effects and promoted apoptosis, increasing cleaved caspase-3 by 1.5-fold and TUNEL staining by 1.9-fold (p<0.01). Flow cytometry confirmed a significant increase in early and late apoptotic cells in the SsnB group. SsnB also increased ceramide (C18, C20, C22, C24) levels (1.3- to 2.5-fold, p<0.01) while reducing PMA-induced S1P and C1P increases (p<0.01).

  1. Why did the authors select Sparstolonin B (SsnB) for this study. What is rationale behind selecting this natural compound.

We selected Sparstolonin B (SsnB) for our study due to its documented anti-inflammatory and anti-proliferative properties, which align with our research objectives in targeting colorectal cancer cells. SsnB, a natural compound derived from the plant Sparganium stoloniferum, has been shown to modulate Toll-like receptor (TLR) signaling pathways, particularly TLR2 and TLR4, leading to reduced inflammatory responses. This modulation is significant because TLR signaling is implicated in cancer progression and inflammation [15]. Furthermore, SsnB has demonstrated the ability to inhibit cell proliferation and induce apoptosis in various cancer cell lines. For instance, studies have reported that SsnB induces cell cycle arrest and promotes apoptotic cell death in neuroblastoma cells through the generation of reactive oxygen species [4]. Additionally, SsnB has been observed to suppress tumor growth and induce apoptosis in prostate cancer cells by inhibiting the PI3K/AKT pathway via reactive oxygen species accumulation [27]. Given these findings, we hypothesized that SsnB could exert similar anti-cancer effects on colorectal cancer cells by modulating inflammatory pathways and inhibiting cell proliferation. Our study aimed to explore these potential mechanisms, providing further insight into SsnB's applicability as a therapeutic agent in colorectal cancer.

References:

  1. Liang Q, Wu Q, Jiang J, Duan J, Wang C, Smith MD, Lu H, Wang Q, Nagarkatti P, Fan D. Characterization of sparstolonin B, a Chinese herb-derived compound, as a selective Toll-like receptor antagonist with potent anti-inflammatory properties. J Biol Chem. 2011 Jul 29;286(30):26470-9. doi: 10.1074/jbc.M111.227934. Epub 2011 Jun 10. PMID: 21665946; PMCID: PMC3143611.
  2. Kumar A, Fan D, Dipette DJ, Singh US. Sparstolonin B, a novel plant derived compound, arrests cell cycle and induces apoptosis in N-myc amplified and N-myc nonamplified neuroblastoma cells. PLoS One. 2014 May 1;9(5):e96343. doi: 10.1371/journal.pone.0096343. Erratum in: PLoS One. 2016 Jul 6;11(7):e0159082. doi: 10.1371/journal.pone.0159082. PMID: 24788776; PMCID: PMC4006872.
  3. Liu S, Hu J, Shi C, Sun L, Yan W, Song Y. Sparstolonin B exerts beneficial effects on prostate cancer by acting on the reactive oxygen species-mediated PI3K/AKT pathway. J Cell Mol Med. 2021 Jun;25(12):5511-5524. doi: 10.1111/jcmm.16560. Epub 2021 May 5. PMID: 33951324; PMCID: PMC8184693.
  4. Why did the authors use Phorbol 12-myristate 13-acetate (PMA). There are other cancer promoters.

We sincerely appreciate the reviewer’s insightful comments regarding our choice of PMA as the tumor promoter in our study. To clarify our rationale, we have now expanded the discussion section to include a detailed explanation of PMA’s well-documented role in stimulating proliferation, inflammation, and TLR signaling pathways, particularly in colorectal cancer models. Specifically, we have added the following text to the revised discussion:

"We selected PMA as the tumor promoter and inflammatory activator in our study due to its well-documented role in stimulating proliferation, inflammation, and TLR signaling pathways, particularly in colorectal cancer models. PMA effectively mimics the tumor-promoting effects of chronic inflammation, which is a key driver in colorectal cancer progression [24]. PMA has been shown to induce the expression of Toll-like receptors (TLRs), particularly TLR2 and TLR4, in various cell types. For instance, a study demonstrated that PMA-mediated differentiation of HL-60 cells into macrophage-like cells resulted in a significant increase in TLR2 expression and, to a lesser extent, TLR4 expression. This upregulation was found to be dependent on protein kinase C activation [25]. PMA is frequently used in cancer studies, including colorectal cancer models, to study pro-inflammatory and pro-tumorigenic signaling cascades [26]."

We believe this revision strengthens the justification for our choice of PMA and aligns with the reviewer’s valuable feedback. We sincerely appreciate the constructive comments, which have helped us improve the clarity and depth of our discussion.

References

  1. Kundu JK, Surh YJ. Inflammation: gearing the journey to cancer. Mutat Res. 2008 Jul-Aug;659(1-2):15-30. doi: 10.1016/j.mrrev.2008.03.002. Epub 2008 Mar 16. PMID: 18485806.; Griner EM, Kazanietz MG. Protein kinase C and other diacylglycerol effectors in cancer. Nat Rev Cancer. 2007 Apr;7(4):281-94. doi: 10.1038/nrc2110. PMID: 17384583.
  2. Li C, Wang Y, Gao L, Zhang J, Shao J, Wang S, Feng W, Wang X, Li M, Chang Z. Expression of toll-like receptors 2 and 4 and CD14 during differentiation of HL-60 cells induced by phorbol 12-myristate 13-acetate and 1 alpha, 25-dihydroxy-vitamin D(3). Cell Growth Differ. 2002 Jan;13(1):27-38. PMID: 11801529.
  3. Marcuello M, Mayol X, Felipe-Fumero E, Costa J, López-Hierro L, Salvans S, Alonso S, Pascual M, Grande L, Pera M. Modulation of the colon cancer cell phenotype by pro-inflammatory macrophages: A preclinical model of sur-gery-associated inflammation and tumor recurrence. PLoS One. 2018 Feb 20;13(2):e0192958. doi: 10.1371/journal.pone.0192958. PMID: 29462209; PMCID: PMC5819803.
  4. Many experiments use a single dose (25 μM SsnB) and time point (12 hours). Dose-response curves for TLR signaling and sphingolipid changes, as well as longer incubation periods (24–48 hours), would provide deeper insights into efficacy and toxicity thresholds.

We appreciate the reviewer’s suggestion regarding the inclusion of dose-response curves for TLR signaling and sphingolipid changes, as well as longer incubation periods (24–48 hours). Our viability studies already encompass a wide range of SsnB concentrations (3.125–50 μM) and different incubation times (12, 16, and 18 hours). The results indicate that 25 and 50 μM SsnB significantly reduced cell viability at all tested time points in HCT-116 cells, with the most substantial reduction observed at 18 hours. In BJ cells, only 50 μM SsnB significantly reduced viability, whereas 25 μM had no notable cytotoxic effect. These findings guided our choice of 25 μM SsnB for subsequent experiments.

The decision to use a 12-hour time point was based on the balance between efficacy and cell viability. At this time point, HCT-116 cells showed approximately 50% viability, which allows for measurable biological effects without excessive cell death that could confound downstream analyses. PCNA fluorescence staining and ELISA results confirmed that SsnB significantly reduced proliferation in HCT-116 cells after 12 hours, further supporting our selected time frame. Extending beyond this time frame may lead to excessive cell death, making it difficult to differentiate between cytotoxicity and specific biological effects. However, we acknowledge the reviewer’s valuable suggestions and will consider performing additional dose-response analyses while extending time points in future studies to further elucidate the long-term efficacy and toxicity thresholds of SsnB.

  1. The study relies solely on HCT-116 cells and BJ fibroblasts. To strengthen generalizability, additional colorectal cancer cell lines (e.g., SW480, Caco-2) and non-cancerous epithelial cells (e.g., NCM460) should be included to confirm that observed effects are not cell-line-specific.

We appreciate the reviewer’s insightful suggestion to include additional colorectal cancer cell lines (e.g., SW480, Caco-2) and non-cancerous epithelial cells (e.g., NCM460) to enhance the generalizability of our findings. We fully acknowledge the importance of validating our results across multiple cell lines to rule out cell-line-specific effects. Due to financial and resource constraints, we were unable to conduct experiments in additional cell lines at this time. However, we believe our study provides meaningful insights by including both cancerous (HCT-116) and non-cancerous (BJ fibroblasts) cells, which allowed us to assess differential responses to SsnB.

We recognize the value of expanding our investigation to other colorectal cancer and non-cancerous epithelial cell lines and consider this an important direction for future studies as additional resources become available.

  1. The authors should indicate future perspectives of their results.

We appreciate the insightful suggestion regarding future perspectives. In response, we have included a discussion on potential directions for further research as written below.

 Prospective studies should explore the effects of SsnB in additional colorectal cancer and non-cancerous epithelial cell lines to confirm broader applicability. Further mechanistic investigations are needed to elucidate its role in apoptosis and inflammatory signaling pathways. In vivo studies will be crucial to assess its therapeutic efficacy, safety, and potential impact on the tumor microenvironment. Additionally, evaluating SsnB in combination with chemotherapeutic agents may provide insights into its potential to enhance treatment efficacy or overcome drug resistance. Finally, understanding its role in sphingolipid metabolism may uncover novel therapeutic targets. These future directions will help establish SsnB’s potential as a candidate for colorectal cancer treatment.

Round 2

Reviewer 1 Report

Comments and Suggestions for Authors

The authors have addressed all the queries raised by honorable reviewers and the manuscript is now much improved, especially the concern regarding the concentration of DMSO. I accept the article in its present form for publication.

Reviewer 2 Report

Comments and Suggestions for Authors

Pharmaceuticals (Manuscript ID: pharmaceuticals-3478104), Comments to the Authors:

Title: Sparstolonin B Suppresses Proliferation, Modulates Toll-Like Receptor Signaling and Inflammatory Pathways in Human Colorectal Cancer Cells

Comments

After reading the authors response to my comments, I think the revised paper can be accepted for publication.